# How animal ethics committees make decisions – a scoping review of empirical studies

**Aoife Milford**[1]*, **Eva De Clercq**[1], **Edwin Louis-Maerten**[1], **Lester D. Geneviève**[1], **Bernice S. Elger**[1,2]

1 Institute for Biomedical Ethics, University of Basel, Basel, Switzerland, 2 Center of Legal Medicine, Faculty of Medicine, University of Geneva, Geneva, Switzerland

* aoife.milford@unibas.ch

## Abstract

### Objectives

The aim of the scoping review is to explore the decision-making process for the evaluation of animal research proposals within Animal Ethics Committees (AEC) and Institutional Animal Care and Use Committees (IACUC), and to critically summarize the available empirical literature on the different factors influencing, or likely to influence, decision-making by AECs when evaluating animal research proposals.

### Methods

A systematic search of empirical literature published between 01.12.2012 and 03.06.2024 in PubMed, Scopus, and Web of Science, was performed.

### Results

Twelve papers were included in the final results, four of which were quantitative, five qualitative, and three were mixed methods. Qualitative content analysis revealed deficits in the assessment of the 3Rs (Replacement, Reduction or Refinement) or the weighing of harms and benefits. Factors related to the review process, applicants, and committees were found to influence this process.

### Conclusion

The findings prompt pragmatic strategies to improve the decision making process of Animal ethics committees.

### Registration

The protocol for this review was registered with Open Science Framework (OSF) with the following DOI: https://doi.org/10.17605/OSF.IO/GZJMB

**Data availability statement:** All relevant data are within the manuscript and its Supporting Information files.

**Funding:** The authors acknowledge the support of the Swiss National Science Foundation (SNSF National Research Program (NRP) - 79 Advancing 3R – Animals, Research and Society, grant number 206432 and 206458) in funding the APC The funders had no role in study design, data collection and analysis, decision to publish, or preparation of the manuscript.

**Competing interests:** The authors have declared that no competing interests exist.

## Introduction

In 2020, an estimated 8 million animals (mostly vertebrates) were used for research and testing purposes in the EU [1] with more than 100 million vertebrates used globally [2]. Animal use in research results in both ethical and political challenges. Although there are those who oppose all animal experimentation, overall public acceptance is contingent on adherence to good welfare standards and the restriction of animal use to designated purposes [3].

One way to uphold humane use of animals for experimentation is via the implementation of the 3R principles (Replacement, Reduction, Refinement) proposed by Russel and Burch in 1959 [4]. This widely adopted framework aims to promote animal welfare by: replacing animals wherever feasible with another means of experimentation; reducing the number of animals used to the minimal needed to obtain scientifically valid results; and to refine experiments to avoid unnecessary harm to animals [5]. However, this framework does not provide guidance on which experiments should take place. In this regard, Beauchamp and DeGrazia [6] proposed six principles of morally justified research, which need to be satisfied for an animal experiment to be conducted. These are: 1) no alternative method to animal testing exists, 2) an expected net benefit to humans, 3) the experiment provides sufficient value to justify the harm caused to animals, 4) animals are not subjected to unnecessary harm, 5) basic needs of animals are met, and 6) there are upper limits to the harm allowable in an experiment. A significant part of these principles is the weighing of expected harms against potential research benefits, a process often referred to as a harm-benefit analysis (HBA). Even if these conditions are met, scientifically unsound research amounts to unnecessary harm [7] since the design features of an experiment affect the validity and reproducibility of its results [8]. For this reason, some researchers argued that an assessment of the scientific validity of animal research protocols should be an essential part of an HBA [8].

Different decision-making entities are tasked to carry out evaluations of animal research protocols, such as Animal Ethics Committees (AEC) or Institutional Animal Care and Use Committees (IACUC). These committees can be made up of veterinarians, practicing scientific researchers, lawyers, ethicists, and sometimes, lay members of the public. The composition of the committees varies depending on context and regulation. In the US, for example, IACUCs must include at least one veterinarian, one scientist, one non-scientist (ethicist, lawyer, or member of the clergy), and one member unaffiliated with the institution [9]. In contrast, EU legislation requires the presence of technical experts, animal welfare experts, and scientific researchers [10]. The framework for reviewing animal experiments also varies from country to country. Some countries, such as Canada, do not legislate for animal experiments directly but nonetheless have a self-regulating oversight system in the form of IACUCs, which incorporate frameworks such as HBAs and 3R [11]. In Japan, the 1973 Act on Animal Welfare and Management of Animals [12] regulates how experiments are carried out while deferring to self-regulated IACUCs at an institutional level. In the US, the Animal Welfare Act 1966 requires 'review' of animal experiments by an IACUC. Within this act, scientific evaluation of experiments is not required unless related to animal welfare. With regard to the EU, countries are subject to the 2010/63/EU directive [13]. In particular, article 36 requires an assessment of the 3Rs, a HBA, and an evaluation of experiment design by a competent authority designated by the state. A similar approach as in the EU is adopted by New Zealand [14] and Australia [15] whose legislation also requires an assessment of the 3Rs, HBA, and experimental design for scientific validity. Given these legal differences, it is understandable that the decision-making process differs between countries. Even between EU countries, there are significant differences in how EU legislation is being implemented [16,17].

Beyond national and regional differences, variations in decision-making have been noted between different committees in the same country. Poor reliability between

committees was identified in final decisions made [18] and in assessments of costs to animals, expected benefits, and recommended modifications [19]. Given these inconsistencies, it is hardly surprising that public trust in regulatory systems around animal testing is low [3].

In 2012, a literature review charted some of the issues impacting AECs' decision-making processes [20]. Key issues noted were: membership composition influencing the effectiveness of committees; narrow ethical views within committees; high rates of approval of protocols; and poor reliability for the justification of animal use. The introduction of European directive 2010/63/EU shortly preceded this 2012 review. This directive was followed by further updates to legislation in other countries. For example, the revision of the Australian Code of Care for the Use of Animals for Scientific Purposes (2013), and the US revision of the Policy on Humane Care and Use of Laboratory animals (2015). These updates require a re-evaluation of the decision-making process. Another recent systematic review focused on methods that have been used to evaluate IACUCs in the US [21]. There is some overlap in references used between this review and our current review although the research question is different. Our current review focusses on the decision making process of animal ethics committees and is not limited to US studies.

The overall objective of the scoping review is to explore the decision-making process for the evaluation of animal research proposals within AECs and IACUCs especially in light of legislative changes over the last decade. More specifically, it has the following aims: (1) to gain a deeper understanding of the decision-making process of AECs when evaluating animal research proposals, and (2) to critically summarize the available empirical literature on the different factors influencing, or likely to influence, decision-making by AECs when ethically evaluating animal research proposals.

## Method

This scoping review adheres to the Preferred Reporting Items for Systematic Reviews and Meta-Analyses for scoping reviews (PRISMA-ScR) [22]. Its protocol was registered on the Open Science Framework (OSF) with the following DOI: https://doi.org/10.17605/OSF.IO/GZJMB.

The scoping review approach was chosen to: (1) identify the types of empirical evidence available on decision-making processes in AECs/IACUCs, (2) to catalogue the key characteristics or factors related to decision-making, and (3) to identify and analyse knowledge gaps [23]. Following JBI methodology for scoping reviews, the Population-Concept-Context (PCC) framework was used to guide the focus of the review [24]. To keep the search as broad as possible, context (animal experimentation) was omitted as an exclusion criterion to prevent overlap with the population (animal ethics committees). The search strategy was first developed for PubMed and then adapted for Scopus and Web of Science (core collection). The Medical Subject Headings (MeSH) and search terms were identified based on extensive background reading on decision-making and animal ethics committees. Only search terms in English were included. The search strategy was checked and validated by an information specialist of the University of Basel library. The search was performed first on 11.05.2023 and then updated on 03.06.2024 (dd.mm.yyyy).

As part of the inclusion criteria, the date range for publication of sources was 01.12.2012 to 03.06.2024 to include all publications since the time of the last review on the topic [25]. Only empirical studies published in peer reviewed journals were included in the languages known to the authors (English, French, German and Italian), there was no limitation on the type of empirical data to be collected. Theoretical papers, reviews, books, editorials, and book chapters were excluded as were articles that did not separate data on

animal ethics committees from human ethics committees or institutional review boards. Papers were also excluded if they did not include data about animal ethics committees making decisions about experiments in biomedical research, *i.e.,* experiments for education were not included.

The results were collated using EndNote X9 software and then transferred to Covidence, an online software used to remove duplicates and facilitate the screening process [26]. Title-and-abstracts and full texts screening were performed independently by two review authors (AM and ELM) with a third author (LDG) acting as arbitrator in the case of disagreement. Once the full texts were screened, a citation and reference search was carried out on all included reports. Independently extracted data were collated using a data charting form including general study characteristics: author names, year of publication, country of origin, journal name, study design, study participants and study aim. To meet the objectives of this review, we also collected information on the factors influencing the decision-making process or details related to the *modus operandi* of animal ethics committees. Data extraction was done by AM and ELM and reviewed by LDG and EDC to ensure accuracy.

Basic qualitative content analysis was used to map and identify key characteristics in the diverse data [27]. This process identifies the following procedure as appropriate for scoping reviews. To begin authors AM and ELM immersed in the data with several readings of the included reports. During the charting and analysis process, an inductive framework was developed. Initial thoughts and notes were recorded for each source to create a coding framework. Extracted information was then organized within this framework. The framework was then reviewed to develop overarching categories within the data.

## Results

Nine hundred and sixty publications were obtained by database searches, in which 298 duplicates were found and removed. The remaining 662 title-and-abstracts were then independently screened by AM and ELM. This resulted in the inclusion of 95 studies for full-text screening by AM and ELM. Eighty three studies were excluded for the following reasons; irrelevant outcomes-they did not address the topic of decision making in animal ethics committees (n = 8); irrelevant language-it was published in a language unknown to the authors (n = 1); irrelevant population- the data collected in the study was not collected from animal ethics committees or animal ethics committee members specifically looking at animal research protocols either through self-report method or through analysis of documents related to committees (n = 3); or irrelevant study design- the papers did not include empirical evidence (n = 71). This resulted in 12 eligible studies available for data charting. Citation and reference searching for the eligible studies resulted in no additional sources. The process of study selection is outlined below (Figure 1).

To present our findings, we will start with providing some general study characteristics (details of included studies are outlined in Table 1). Of the 12 included papers, 4 were quantitative, 5 qualitative and 3 mixed methods studies. Seven relied on self-reported data, while the other five consisted of documents generated by committees or transcripts of committee discussions. Six of the retrieved literature originated in the US and of those, four had the same first author. Of the studies involving a European cohort, three of the five were from Sweden. Most papers (n = 9) were published in animal specific journals, only three were published in general bioethics or research ethics journals. In a next step, we will discuss two main themes and various subthemes: (1) principles guiding the review process and (2) factors influencing the review process. Distribution of codes and the studies in which they were present is shown below (Table 2).

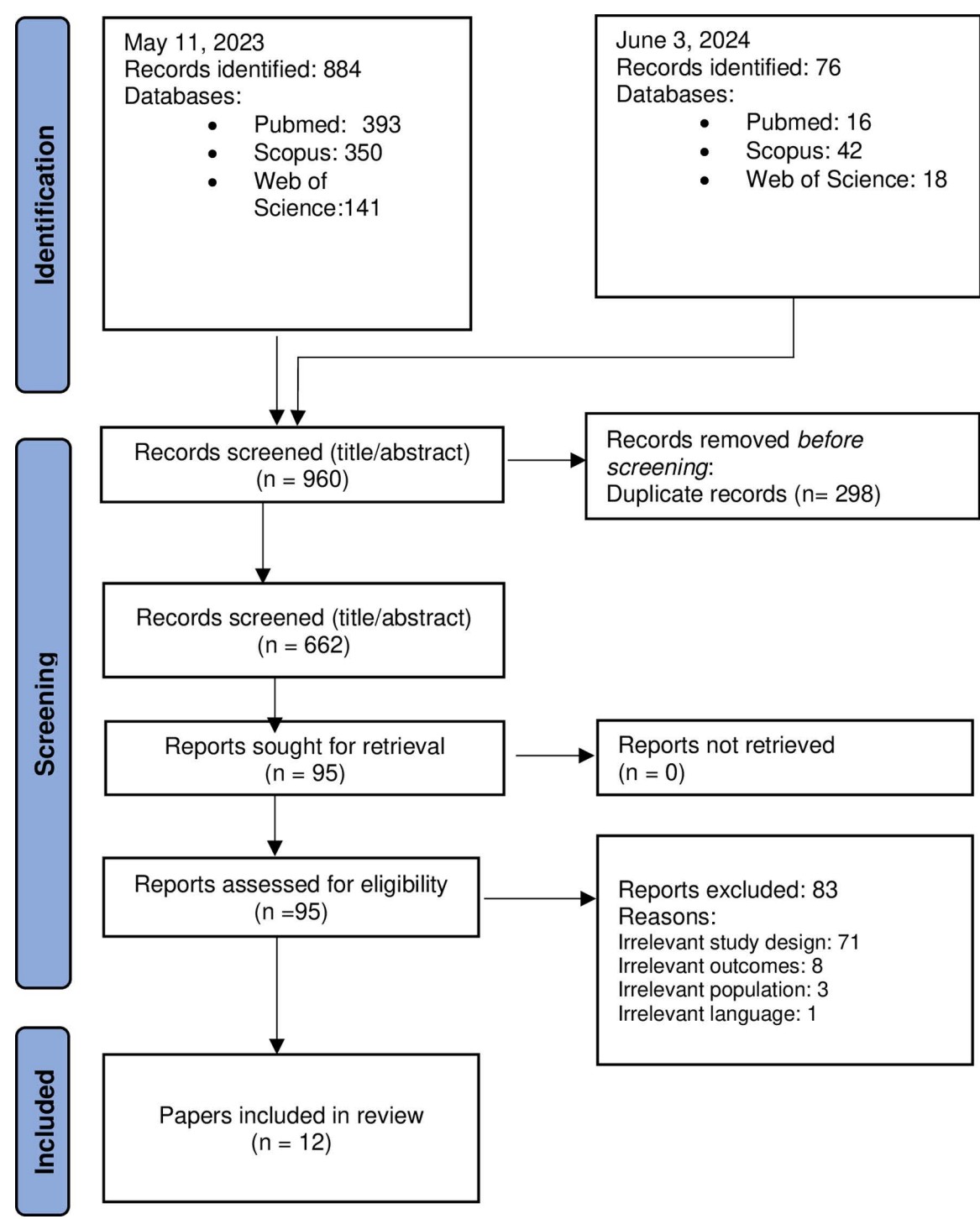

**Fig 1. PRISMA diagram systematic search and selection process.**

**Table 1. Summary of included publications (N = 12).**

| Study | First author | Title | Country | Journal | Publication Year | Type of data | Study design | Sample |
|---|---|---|---|---|---|---|---|---|
| 1 | Colman [28] | Marmosets: Welfare, Ethical Use, and IACUC/Regulatory Considerations | USA, Japan, Netherlands, UK | ILAR | 2020 | mixed | Survey | 18 institutions (Institutional Animal Care and Use Committees and Animal Ethics Committees) |
| 2 | Hansen [29] | Analysis of Animal Research Ethics Committee Membership at American Institutions | USA | Animals | 2012 | Quantitative | Roster collection | 21 Institutional Animal Care and Use Committees |
| 3 | Jorgensen [30] | Reviewing the Review: A Pilot Study of the Ethical Review Process of Animal Research in Sweden | Sweden | Animals | 2021 | Qualitative | application and decision documents | 18 documents from 6 different Animal Ethics Committees |
| 4 | Ormandy [31] | Animal Research, Accountability, Openness and Public Engagement: Report from an International Expert Forum | Australia, Canada, Germany, New Zealand, Sweden, UK, USA | Animals | 2019 | Qualitative | Focus group | 11 participants with expertise in governance of animal research in their country of residence |
| 5 | Ringblom [32] | Assigning Ethical Weights to Clinical Signs Observed During Toxicity Testing | Sweden | Altex | 2017 | Quantitative | Interviews | 47 Animal Ethics Committee members from different Institutions |
| 6 | Silverman [33] | A self-assessment survey of the Institutional Animal care and use committee, Part 1: animal welfare and protocol compliance | USA | Lab Animal | 2012 | Quantitative | Survey | 768 Institutional Animal Care and Use Committee members from different institutions |
| 7 | Silverman [34] | A self-assessment survey of the Institutional Animal care and use committee, Part 2: structure and organizational functions | USA | Lab Animal | 2012 | Quantitative | Survey | 1007 Institutional Animal Care and Use Committee members from different institutions |
| 8 | Silverman [35] | Decision Making and the IACUC: Part 1— Protocol Information Discussed at Full-Committee Reviews | USA | Journal of the American Association for Laboratory Animal Science | 2015 | qualitative | focus group | 87 discussion transcripts from 10 Institutional Animal Care and Use Committees |
| 9 | Silverman [36] | Factors Influencing IACUC Decision Making: Who Leads the Discussions? | USA | Journal of Empirical Research on Human Research Ethics | 2017 | Mixed | Focus group | 87 discussion transcripts from 10 Institutional Animal Care and Use Committees |
| 10 | Tjärnström [37] | Emotions and Ethical Decision-Making in Animal Ethics Committees | Sweden | Animals | 2018 | Mixed | Interview and survey | 74 Animal Ethics Committee members from different institutions |
| 11 | Vardigans [38] | Breaking barriers to ethical research: An analysis of the effectiveness of nonhuman animal research approval in Canada | Canada | Accountability in Research | 2019 | Qualitative | Animal use protocol forms | 14 animal use protocol forms from different institutions |
| 12 | Waltz [39] | Mission Creep or Mission Lapse? Scientific Review in Research Oversight | USA | AJOB Empirical Bioethics | 2022 | Qualitative | Interviews | 45 prior or current Institutional Animal Care and Use Committee and Institutional Review Board members from different institutions |

## Principles guiding the review process

### a) 3R assessment

Six studies dealt with the topic of 3R, revealing an overall focus on reduction and refinement over replacement. Five studies explored reduction, revealing discrepancies between

**Table 2. Distribution of themes, subthemes and codes.**

| Theme | Sub theme | Code | Studies included (first author, publication year) |
|---|---|---|---|
| Principles guiding the review process | 3R | Reduction | Colman 2020; Jorgensen 2021; Silverman 2021; Silverman 2012a; Vardigans 2019 |
| | | Refinement | Colman 2020; Jorgensen 2021; Vardigans 2019 |
| | | Replacement | Jorgensen 2021; Silverman 2012a; Silverman 2015; Vardigans 2019 |
| | | Reduction and refinement trade-off | Ringblom 2017 |
| | Harm benefit analysis | Harms to animals | Colman 2020; Jorgensen 2021; Ringblom 2017; Silverman 2015; Silverman 2017 |
| | | Social benefit | Silverman 2015; Silverman 2017; Vardigans 2019; Waltz 2022 |
| | | Weighing of harms and benefits | Jorgensen 2021; Silverman 2012a; Tjärnström 2018; Waltz 2022 |
| | | Lack of guidance | Colman 2020; Jorgensen 2021; Silverman 2012a |
| | Scientific Validity | Appropriateness of animal model | Colman2020; Silverman 2012a; Silverman 2015 |
| | | Scientific quality | Waltz 2022 |
| Factors influencing the review process | Factors related to the review process | Fragmentation of the review | Waltz 2022 |
| | | Independence of the review | Ormandy 2019; Silverman 2012a; Silverman 2012b |
| | Factors related to the applicants | Credentials of applicants | Silverman 2012b; Waltz 2022 |
| | Factors related to the committee | Independence of the review | Ormandy 2019; Silverman 2012a; Silverman 2012b |
| | | Committee composition | Hansen 2012; Ormandy 2019 |
| | | Competence of committee members | Ormandy 2019; Silverman 2012a; Silverman 2017 |
| | | Group dynamics and leadership | Ormandy 2019; Silverman 2012b; Silverman 2017; Tjärnström 2018 |
| | | Diversity of values amongst committee members | Ringblom 2017; Silverman 2012a; Silverman 2017; Tjärnström 2018 |

self-reported practices and analysis of documentation [28,30, 33, 35,38]. While most committee members claimed to ensure appropriate numbers of animals are used [28,33], evidence from discussion transcripts [35], or review forms [30] revealed that reduction assessments were overlooked, insufficient or narrowly focused on animal numbers of an experiment or statistical methods rather than any other method of reduction such as the use of embryo transfer use in breeding programs or animal/ tissue/ data sharing.

Half of the retrieved studies dealt with the topic of refinement and found that refinement is an important topic for committees [28,30,31,33,35,38]. This was particularly true for non-human primates with frequent, in depth, discussions on animal welfare issues such as social housing, cage structure, enrichment, anaesthesia, or sanitation procedures [28]. A trade-off

between reduction and refinement was observed; some committee members prioritized reducing animal distress even if it required more animals, though about a third of participants rejected this compromise [32].

Replacement appeared in four studies, again with discrepancies between self-reported data analysis of actual discussion transcripts. One study showed that committee members claimed to carry out thorough searches for alternatives as part of their review [33] but other studies relying on analysis of completed applications, decision documents and application forms revealed sporadic discussion or evaluation of replacement [30,35,38].

**b) Justification and Harm-benefit analysis**

Eight studies examined the HBA, showing that harms were, in general, discussed more frequently and in more detail than benefits [28,30,32,35–39]. Commonly discussed harms included procedures to be performed on animals and animal death [28,35,36], while topics like painful or distressing procedures or the upper limit of suffering were less often addressed [30,36]. Only five studies dealt with benefit analysis revealing this as one of the least mentioned topics for discussion being rarely requested in application forms ahead of review [35,36,38,39]. Some committee members reported being specifically asked not to review social value but to focus solely on harm mitigation as part of the review process [31].

Deficits in weighing harms against benefits were found with some committee members feeling confident the process whilst others believed that decisions were centred around technical improvements [37]. Analysis of decision documents showed unclear or vague reasoning in the decisions made with no clear outline of the rationale behind decisions [30]. Three studies underscored the need for better review guidance [28,30,33]. Two studies called for more clear, detailed and prescriptive guidance in order to carry out a review [30,33] whereas the third recommended updating standard operating procedures with specific information for certain species such as marmosets [28].

**c) Assessment of scientific validity**

Only one study dealt with the topic of assessing the research design in terms of validity or quality finding that such assessments of validity were rare in the US where there is a lack of clear guidance on whether they are required to do so [39]. Three studies examined species appropriateness in reviews [28,33,35]. Detailed justification for the choice of non-human primates was reported in one study [28]. This contrasted with another study which found that although species justification was a common theme for discussion, it was rarely discussed in detail [35]. Another study found that while rodents were viewed as less sentient, committee members believed all protocols received equal scrutiny [33].

## Factors influencing the review process

The next section deals with a range of factors that may influence the review process. This included factors related to: (i) the fragmented structure of the ethical review process; (ii) applicants seeking protocol approval; and (iii) the committee evaluating animal research protocols.

**A fragmented process.** One U.S. study noted the fragmentation of review processes, with 3R, HBA, and validity assessments sometimes conducted by separate, non-communicating bodies. Prior funding approval often impacted both scientific merit and social value evaluations [39].

**Applicant-related factor(s).** Five studies explored applicants' skills and expertise [33,35,36,39]. Two studies noted that while applicants' credentials were frequently raised, they were seldom dealt with in detail with few comments made on this topic per discussion [35,36].

Committee members in one study revealed that trust in the expertise of applicants results in less scrutiny of the quality of research [39]. In contrast, survey data in another study [34] claimed that applicants' credentials had no effect on review quality.

**Committee-related factors.** A range of committee-related factors were found: (1) affiliation/independence of committee members (2) composition of the committees (3) expertise of the committee members (4) group dynamics (5) variances between types of committee members.

Three studies discussed the independence of the review process [31,33,34]. Review committees in some countries are affiliated with research institutions whereas in other countries they are independent. Survey data revealed that most committee members felt that affiliated review committees can focus on local needs [34]. However, another study [33], using the same cohort of participants, revealed that some committee members were involved to ensure their research aims were not obstructed by review committees.

Two studies dealt with the committee composition revealing an imbalance favouring biomedical researchers and veterinarians [29,31]. Participants from an international forum discussion emphasised the importance of a diverse committee to reduce bias and allow consideration of broader perspectives [31].

The expertise available in committees was explored in four studies [31,33–35]. Concerns were raised about deficits in Biostatistics [35], ethics or bioethics [34,40], and up to date regulations [34,40]. One study [33] reported that most IACUC members believe that their members are adequately trained to review protocols and, advice could be solicited from a non-IACUC specialist when needed. Survey data from two studies, reported that most committee members felt they were conscientious, active, and well prepared members of the committees [33,34].

Four studies described the role of group dynamics in AEC/IACUC meetings [31,34,36,37]. Some committee members in one study [34] reported that veterinarians and scientists overly influenced discussions and that chairpersons guided the direction of discussions or deferred their opinion to veterinarians more than other members. Another study found that the presenter of a protocol (most often researchers) influenced discussions by raising topics [36]. By tracking contributions, another study revealed that veterinarians and chairpersons contributed the most comments per person, while scientists, despite making the majority of the comments, contributed less per person compared to other committee members [36]. Variances in perceived ability to influence discussions were reported by certain committee members in one study who felt excluded due to their personalities, lack of knowledge or experience, or emotional appearance [37]. Some interviewees in this study described tensions or a harsh discussion climate, with animal lay members having the least positive experience. Committee members in another study, stressed the importance of good leadership to ensure that all members were included and that they could cope with technical language and details [31].

Four studies identified variances in priorities and viewpoints between types of members in a committee [32,33,36,37]. For instance, one study [32] found that researchers and members of animal welfare groups rated clinical signs of harm at a higher ethical cost than increasing the number of animals used in an experiment than politically nominated lay persons. Another study found that although animals evoked more empathy than those who would benefit from research (*e.g.*, patients), overall, political laypersons were more empathetic to patients, researchers equally to patients and animals, and animal technicians primarily empathised with animals [37]. The same study found that researchers were more likely to believe that all relevant aspects are considered during ethical evaluation, or that animal testing is justified if it leads to human benefit or that human suffering is more important than animal suffering than other types of members in the committees. Another study [36] found that scientists were

more likely than other groups to discuss the justification of the species used or the numbers of animals used or the skills and competence of the research team than did other members.

## Discussion

This scoping review provides an overview of the empirical literature on the decision-making process in animal ethics committees. The 3R principles, as they were first presented by Russel and Burch, have a hierarchy, in which replacement is given the top priority followed by reduction and refinement [4,5]. In contrast, our results show that refinement is given the most prominent place in discussions and replacement the least [35]. Given that scientists often form the majority of the committee members [29] and that scientists tend to rank refinement as higher than replacement [41], this is not a surprising result. Refinement is also considered by scientists to be the most achievable [41,42]. Additionally, our results show that committees typically have at least one veterinarian who is trained to address refinement techniques and animal welfare needs, and that veterinarians – alongside chairpersons – are contributing the most comments per person [36]. Therefore, discussions of animal research protocols could potentially be stirred towards refinement methods by the latter, at the detriment of replacement or reduction techniques. For other committee members, implementing refinement methods such as improved housing or enrichment may be more straightforward to understand and adopt. It is thus possible, that the availability and familiarity of knowledge surrounding each 'R' influence their weight in the evaluation process.

Reduction on the other hand is more challenging. A common reduction strategy is the use of statistics to ensure the minimum number of animals are used to obtain valid results. This requires knowledge of statistics, which our results show, is not always available to the majority of AEC members [33,40]. Other important means of reduction such as data and tissue sharing through databases such as the SEARCH framework in the US [43] or the 3R blackboard in the EU [44] were not discussed in all our retrieved literature. Similarly, other authors have demonstrated that knowledge of replacement methods is lacking and there are some concerns about the validity of replacement methods [45] or fears about publishers preferring animal studies [46], which may lead scientists to neglect replacement in favor of refinement or reduction. This is especially important given that the majority of AEC members are biomedical researchers [29]. Nonetheless, it is possible that animal methods bias (a preference for animal methods when alternatives are available) and scientific inertia or tradition are reasons why replacement methods are less considered [47,48].

HBAs are the linchpin for the justification of animal research, yet it is evident that there are a range of challenges with the implementation of an HBA. Two such challenges were found in our results. First, assessing benefits in comparison to harms is a subjective task, which is considered difficult and frustrating to carry out by committee members [37,49]. One cited reason in the retrieved literature was a lack of clear guidance on how to carry out a HBA [30]. The EU directive 2010/63 provides detailed schema to assess harms, whereas the legislation does not provide any schema to assess the benefits of research beyond confining research to certain purposes [17,50]. Similarly, US regulations provide only minimal guidance to ensure research has social or scientific value [39]. It is, therefore, not a surprise that our results which were predominantly from the US or Europe, demonstrate that harms were discussed more frequently and in more detail than benefits by committee members, and that the weighing of harms against benefits was even more sparse [30,35,36,39]. Second, our results show that the benefits of research are often optimistically assumed and not analyzed in detail [39]. Taking into account how few projects are considered to satisfy Bateson's criteria for conducting an HBA retrospectively, this optimism and confidence in research is hard to justify [51]. One proposed solution is to diversify the committees' composition [52]. At present only

technical experts; animal welfare experts (including veterinarians); and scientific researchers are required for committees within the EU [49]. Scientists, however, are more likely to be positively biased towards research than other groups, such as animal activists [10]. Another potential challenge with HBAs is that they represent a narrow consequentialist approach to ethics [49]. Members of the public and key animal ethicists, on the other hand, have a range of ethical positions [53–56]. It is arguable that this may present a source of conflict within committees as to how to approach and conduct a HBA. Where these analyses are conducted using an ethical framework not supported by either ethicists or the public, it is possible that the review process may not be trusted. This may be one reason why evidence points to a lack of public trust in the review process [3].

Our review demonstrated that although the validity of using certain animal models is considered important [28,33,35], the scientific validity of an experiment was less likely to be discussed by committee members [39]. This holds great significance given the emerging evidence of poor reproducibility or translatability of completed animal research in recent years due to poor experimental design [57,58]. The strength of internal validity, reported in protocols submitted for ethical review, correlates with the strength of internal validity reported in publications [59]. This demonstrates that the quality of a research proposal cannot be assumed at ethical review stage and highlights the importance of assessing the scientific validity prior to commencing a project to prevent poorly designed studies going forward. Not only is there minimal benefit to invalid results, but invalid results also represent a waste of resources as well as unnecessary suffering of experimental animals. Despite this, several countries (including the US and Canada) do not require an assessment of scientific validity as part of their review process. For this reason, it is recommended that the HBA should only take place after an in depth assessment of the validity of a research protocol [8].

Our results demonstrated that the review process can be influenced by factors related to the nature of the review process, the applicants, or the members of the committee. The composition of AECs or IACUCs can be imbalanced in favor of researchers and veterinarians which can affect: (i) the expertise available and focus of the review; (ii) the independence of the review; (iii) the empowerment of all members. Although expertise in animal research can be useful in harm reduction, the lack of knowledge in animal research ethics, replacement, or reduction methods can result in a limited ethical review. This also appears to skew ethical review towards harm reduction rather than ethical evaluation. This concern has been echoed by other authors [52,60]. Questions were raised in our results about the independence of reviewers [31,33,34]. Another author explained that this may be a result of the difficulty in recruiting truly independent committee members and cautioned that committee members can be extremely reluctant to raise concerns on ethical grounds with their colleagues [60]. Our review also found that scientists and veterinarians contributed most and tended to lead discussions but that other types of members felt less influential and reported frustration in the process [36,37]. This may be a result of deficient processes for managing moral dilemmas [60]. Ethical review occurs after successful funding applications and our results demonstrate that this can influence reviewers to approve projects [39]. It has been speculated that institutional pressure to approve projects that receive funding may be at play here [60].

In comparison to a previous review on the decision-making process of AECs [25], two issues have remain unchanged. The committee composition still seems to affect its effectiveness and although committees may be more reliable in their analysis of refinement, the justification of animal use remains unclear. There were some other issues that this scoping review had hoped to address. For instance, the ethical codes, regulations and research standards used in practice by AEC members, cultural norms, or the ethical issues deliberated on. These topics however, did not arise in the retrieved literature. Literature on the causes of variations

between different AECs or cultural norms was also missing as was any data about how committee members are selected.

## Recommendations

In light of the issues raised in this review, recommendations could include: 1) New guidance outlining ways to conduct a HBA, and in particular, to scrutinize the benefits of results in detail [30]. 2) Diversification of committees to tackle bias towards animal model; to incorporate a wider set of expertise in reduction and replacement; and to hearken to other voices beyond the researchers and veterinarians [10,52]. 3) Review of the normative framework underlying ethical review to ensure it is functional for multiple stakeholders, scientists, ethicists and members of the public [37]. 4) Training of committee members on the ethical framework underpinning the review process. 5) Training on how committee members with divergent views can positively work together toward an outcome [61]. 6) Scientific validity assessments should be a required integral part of the review process which takes place before the harm benefit analysis [8]. 7) Ethical review should be brought forward to an earlier stage in the planning process to avoid influence from funding approval [60].

## Limitations

The comparison between studies is limited by the legislative requirements in the country of each study resulting in different roles and expectations set for each type of animal ethics committee. The review is also limited by the paucity of results available, which was further reduced by using only English terms for our search. The generalizability of the results is limited due to the narrow range of countries involved. Some of our results drew on studies using self-reported data which can be affected by social desirability or impaired introspective ability. Other results were derived from documents produced by animal ethics committees which may not comprehensively reflect all discussions that took place or the processes involved. Although every effort was made to reduce bias through the systematic search and other methodologies, it is possible that researcher bias played a role in the search strategy or the data extraction process. The scoping review method does not allow drawing conclusions of the effectiveness of a practice overall due to the lack of meta-analysis [27] but this would not have been feasible given the few results available.

## Conclusions

Ethical review of animal protocols is a complex and contentious process on which animals depend for their protection. Through this review we have created a clearer picture of the decision making process and charted some of the deep seated issues surrounding it. At each stage of the evaluation, deficits in the decision-making process were found. The scientific validity of a protocol was often neglected, the 3R assessment tends to focus on refinement rather than reduction or replacement, and the harm benefit analysis often focusses on harms rather than an in –depth weighing of interests. The committees themselves influenced these processes significantly. Committee composition affects the knowledge and expertise available to address key issues, but also the focus of the review. Problems with group dynamics and independence of committees were also discussed. Remedies to address these issues were suggested and although the recommendations given above may be costly, they reflect the deep responsibility for those involved in animal experimentation to protect the animals they work with and ensure responsible research. Further research should delve into the ethical issues debated and the ethical theories informing decision making along with testing measures to address the aforementioned issues.

## Supporting information

**S1. PRISMA Checklist.**
(DOCX)

**S2. Search strategy used for systematic search.**
(DOCX)

## Acknowledgments

The authors thank Christian Appenzeller-Herzog from the University Medical Library of the University of Basel for his assistance in developing the search strategy for this project.

## Author contributions

**Conceptualization:** Aoife Milford, Eva De Clercq.

**Data curation:** Aoife Milford, Edwin Louis-Maerten.

**Formal analysis:** Aoife Milford, Eva De Clercq, Edwin Louis-Maerten, Lester D. Genevieve.

**Funding acquisition:** Bernice S. Elger.

**Methodology:** Aoife Milford, Eva De Clercq.

**Supervision:** Eva De Clercq, Lester D. Genevieve.

**Visualization:** Eva De Clercq.

**Writing – original draft:** Aoife Milford.

**Writing – review & editing:** Aoife Milford, Eva De Clercq, Edwin Louis-Maerten, Lester D. Genevieve, Bernice S. Elger.

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
