## [Decision Letter · Decision Letter 0]

25 Sep 2024

PONE-D-24-34445How do Animal Ethics Committees make decisions – a scoping review of empirical studiesPLOS ONE

Dear Dr. Milford,

Thank you for submitting your manuscript to PLOS ONE. After careful consideration, we feel that it has merit but does not fully meet PLOS ONE’s publication criteria as it currently stands. Therefore, we invite you to submit a revised version of the manuscript that addresses the points raised during the review process.

We look forward to receiving your revised manuscript.

Kind regards,

Jayonta Bhattacharjee

Academic Editor

PLOS ONE

Journal Requirements:

1. When submitting your revision, we need you to address these additional requirements. Please ensure that your manuscript meets PLOS ONE's style requirements, including those for file naming. The PLOS ONE style templates can be found at https://journals.plos.org/plosone/s/file?id=wjVg/PLOSOne_formatting_sample_main_body.pdf and https://journals.plos.org/plosone/s/file?id=ba62/PLOSOne_formatting_sample_title_authors_affiliations.pdf 2. Thank you for stating the following financial disclosure: "The authors acknowledge the support of the Swiss National Science Foundation (SNSF National Research Program (NRP) - 79 Advancing 3R – Animals, Research and Society, grant number 206432 and 206458) in funding the APC". Please state what role the funders took in the study.  If the funders had no role, please state: "The funders had no role in study design, data collection and analysis, decision to publish, or preparation of the manuscript." If this statement is not correct you must amend it as needed. Please include this amended Role of Funder statement in your cover letter; we will change the online submission form on your behalf. 3. We note that your Data Availability Statement is currently as follows: "The data that support the findings of this article are available within the article and its supporting information." Please confirm at this time whether or not your submission contains all raw data required to replicate the results of your study. Authors must share the “minimal data set” for their submission. PLOS defines the minimal data set to consist of the data required to replicate all study findings reported in the article, as well as related metadata and methods (https://journals.plos.org/plosone/s/data-availability#loc-minimal-data-set-definition). For example, authors should submit the following data: - The values behind the means, standard deviations and other measures reported;- The values used to build graphs;- The points extracted from images for analysis. Authors do not need to submit their entire data set if only a portion of the data was used in the reported study. If your submission does not contain these data, please either upload them as Supporting Information files or deposit them to a stable, public repository and provide us with the relevant URLs, DOIs, or accession numbers. For a list of recommended repositories, please see https://journals.plos.org/plosone/s/recommended-repositories. If there are ethical or legal restrictions on sharing a de-identified data set, please explain them in detail (e.g., data contain potentially sensitive information, data are owned by a third-party organization, etc.) and who has imposed them (e.g., an ethics committee). Please also provide contact information for a data access committee, ethics committee, or other institutional body to which data requests may be sent. If data are owned by a third party, please indicate how others may request data access. 4. Please include captions for your Supporting Information files at the end of your manuscript, and update any in-text citations to match accordingly. Please see our Supporting Information guidelines for more information: http://journals.plos.org/plosone/s/supporting-information. 

Reviewers' comments:

Reviewer's Responses to Questions

**Comments to the Author**

1. Is the manuscript technically sound, and do the data support the conclusions?

Reviewer #1: Yes

Reviewer #2: Yes

2. Has the statistical analysis been performed appropriately and rigorously? 

Reviewer #1: Yes

Reviewer #2: N/A

3. Have the authors made all data underlying the findings in their manuscript fully available?

Reviewer #1: Yes

Reviewer #2: Yes

4. Is the manuscript presented in an intelligible fashion and written in standard English?

Reviewer #1: Yes

Reviewer #2: Yes

5. Review Comments to the Author

Reviewer #1: Dear authors,

Thank you for choosing this important topic. Animal ethics is a great topic. I suggest that it be highlighted in scientific studies. Thank you again. Bellow my comments on your manuscript:

Conclusion: Authors wrote the conclusion of their study in abbreviated. I suggest that they rewrite the conclusion in detail, according to the results of their study.

References: excellent, clear, modern and important references.

- Most references without Doi or links.

- Some references are old, such as 1959, 1986 and 1987. I suggest replacing them with modern references.

Reviewer #2: Dear authors,

Thank you for the opportunity to review this interesting manuscript. This paper provides a systemic review of the factors influencing decisions made by Animal Ethics Committees and Institutional Animal Care and Use Committees when evaluating animal research proposals. The authors have elected to make all data available and to provide a registered protocol with Open Science Framework (OSF). Methods are described in a way to allow the systematic review to be reproduced. While a somewhat similar review was conducted in 2020 by Budda and Pritt (Budda, M. L., & Pritt, S. L. (2020). Evaluating IACUCs: Previous research and future directions. Journal of the American Association for Laboratory Animal Science, 59(6), 656-664. https://doi.org/10.30802/AALAS-JAALAS-20-000077), the submitted paper presents a mostly novel objective and content analysis and some important/novel conclusions, though overlap between the two reviews should be addressed and/or discussed in a revised submission (see below). The last review with similar objectives and content analysis to the submitted manuscript was published in 2012 (Varga, O., Sandøe, P., & Olsson, I. A. S. (2012). Assessing the animal ethics review process. In Climate change and sustainable development (pp. 462-467). Wageningen Academic. https://doi.org/10.3920/9789086867530_0073), and the submitted manuscript's systemic review appropriately begins where Varga and colleagues (2012) left off. There are some extremely interesting conclusions presented in the submitted manuscript, especially regarding the 3Rs and the lack of attention given to "replacement" options in current IACUC/AEC discussions/decision-making processes. These results are important and lead to some very pertinent and well-presented recommendations at the end of the paper. Therefore, I do believe this paper should be seen and considered by existing AECs and IACUCs. Overall, the manuscript is well-organized and well-written (save some minor grammatical/spelling/formatting errors described below); however, there are several suggestions that I have included below for consideration in a revised manuscript. These are organized by major suggested revisions (which, in my opinion, MUST be addressed if the manuscript is to be considered for publication) presented first, followed by minor suggested revisions (grammar/spelling/formatting and/or optional edits).

Major revisions:

1. In general, the Results section is too long and can be improved by revising to be more concise. Common themes between articles should be summarized in tables or figures, or using much more concise language in the main text. One suggestion is to move all details about the search and selection process (including Figure 1) to the Methods section, where I believe it fits more appropriately.

2. I am confused about the "one additional paper found through other sources" (lines 141-142). How was this paper found? Why was it not found during the systematic search using described methods? Which paper is this? Throughout the results, it is unclear whether this additional 13th paper is included or not. For example, on lines 154-155, it is stated that "half of the retrieved literature originated in the US..." but half of 13 is 6.5. Please clarify if you mean 6 or 7 papers originated in the US. Moreover, in Figure 1, studies included for review is noted to be n=12. Should this be n=13? Is the method by which this 13th paper was found included in the figure? If not, please include or clarify.

3. Methods state that the search was updated later in 2024 (March or June - date format in Methods needs to be clarified, see below); however, Figure 1 is dated 15th January 2024. Were there no changes implemented after the update? Please clarify the discrepancy.

4. The way in which Table 1 and Figure 2 are presented makes it difficult for the reader to cross-reference the studies included. Studies are dually numbered, 1-13 for Table 1 and 25-37 for the reference list. I believe the table and figure could be combined to limit confusion and present the themes in conjunction with relevant details about each study If combining proves difficult due to the amount of information, I recommend converting Figure 2 into a table (similar to Table 2 in Budda & Pritt, 2020 - referenced above). Presenting Figure 2 in a more explicit manner may help to reduce text in Results section in general as well (see major revision #1).

5. In the Results section, included studies are described as presenting "self-reported data or other data collection methods" (line numbers not provided). Please clarify earlier in the manuscript when describing selection criteria. What types of empirical data were you looking for, specifically? I can deduce that you were looking for either self-reported data, or data that were collected from committee documents. However, this is not explicitly clear to the reader. Were there other types of studies considered?

6. While I appreciate the inclusion of a "Limitations" section, I believe this section is severely lacking. Please discuss other major limitations in this study such as biases in self-reported data, or the potential for missing information from committee documents that may not have covered the full breadth of discussions that took place.

7. Please incorporate Budda & Pritt (2020) - referenced above - into the Discussion section of this paper. There is some overlap that needs to be discussed and justified. As stated above, I do feel that the objective, content analysis, and conclusions of the submitted paper differ enough from Budda & Pritt, 2020 to justify its publication after major revisions; however, I think the overlap can be addressed to strengthen the conclusions presented here and perhaps eliminate some redundancy to make the submitted paper more concise.

Minor revisions:

1. In the abstract, please define all acronyms (i.e., AECs, IACUCs, 3R).

2. Line 43: Please clarify whether "8 million animals" includes all taxa or just vertebrates (as in the second part of the sentence).

3. Lines 114-115: Clarify whether dates provided are mm.dd.yyyy or dd.mm.yyyy format.

4. Line 127: Data should be plural -- "data were collated"

5. Lines 139 and 142: When starting a sentence with a number, spell out the number.

6. Figure 1: References removed for other reasons should be n=0? Current reads "n=".

7. Provide reference to Table 1 in main text.

8. Table 1: Formatting of headings needs to be adjusted to be consistent.

9. Table 1: Clarify for each study if it includes empirical data from multiple IACUCs/AECs or only one. An added column could provide the number of studies summarized in each paper.

10. Figure 1: Remove extra comma for study #s discussing "Independence of review."

11. Figure 1: Please improve consistency of capitalization/sentence case.

12. In the Results (line numbers not provided), it is mentioned that methods of reduction included "animal numbers of an experiment and statistical methods rather than any other method of reduction." Please clarify somewhere what other suggested methods of reduction might be.

13. Section c) Assessment of scientific validity - second line (line numbers not provided): Should read "...US where assessment of scientific validity is not required..."

14. Section c) Committee-related factors - first three lines (line numbers not provided): Unclear why these lines are formatted differently (indented and italicized) than the rest of the section.

15. Sentence in Results section (no line numbers provided) has incorrect grammar. Currently reads "Variances in perceived ability to influence discussions was reported by committee members in one study (35)." Should read "Variance in perceived ability to influence discussions was reported by committee members in one study (35)." OR "Variances in perceived ability to influence discussions were reported by committee members in one study (35)."

16. In Discussion section (no line numbers provided), please clarify what is meant by "animal welfare experts" if different/separate from scientific researchers to clarify claims about scientists' biases toward research.

17. In Recommendation #4 (line numbers not provided), please capitalize the word "Training."

6. PLOS authors have the option to publish the peer review history of their article (what does this mean? ). If published, this will include your full peer review and any attached files.

**Do you want your identity to be public for this peer review?** For information about this choice, including consent withdrawal, please see our Privacy Policy .

Reviewer #1: **Yes: ** Prof.Dr. Khalid C. K. Al-Salhie

Reviewer #2: No

---

## [Author Response · Author response to Decision Letter 0]

11 Nov 2024

Dear Reviewers,

Thank you for your carefully considered comments on how to improve our work. To address the major revisions, we have rewritten the results section, changed figure 2 to a table, updated the PRISMA diagram, added detail to the limitations section, and included a relevant review citation (Budda and Pritt 2020). All changes have been outlined in the attached file (Responses to reviewers) and are visible in the revised manuscript with tracked changes.

We are grateful for the considerable time and attention paid to our paper and we hope we have addressed all issues appropriately.

Best wishes,

Aoife Milford

---

## [Decision Letter · Decision Letter 1]

10 Dec 2024

PONE-D-24-34445R1How do Animal Ethics Committees make decisions – a scoping review of empirical studiesPLOS ONE

Dear Dr. Milford,

Thank you for submitting your manuscript to PLOS ONE. After careful consideration, we feel that it has merit but does not fully meet PLOS ONE’s publication criteria as it currently stands. Therefore, we invite you to submit a revised version of the manuscript that addresses the points raised during the review process.

We look forward to receiving your revised manuscript.

Kind regards,

Jayonta Bhattacharjee

Academic Editor

PLOS ONE

Journal Requirements:

Reviewers' comments:

Reviewer's Responses to Questions

**Comments to the Author**

1. If the authors have adequately addressed your comments raised in a previous round of review and you feel that this manuscript is now acceptable for publication, you may indicate that here to bypass the “Comments to the Author” section, enter your conflict of interest statement in the “Confidential to Editor” section, and submit your "Accept" recommendation.

Reviewer #1: All comments have been addressed

Reviewer #2: (No Response)

2. Is the manuscript technically sound, and do the data support the conclusions?

Reviewer #1: Yes

Reviewer #2: Yes

3. Has the statistical analysis been performed appropriately and rigorously? 

Reviewer #1: Yes

Reviewer #2: N/A

4. Have the authors made all data underlying the findings in their manuscript fully available?

Reviewer #1: Yes

Reviewer #2: Yes

5. Is the manuscript presented in an intelligible fashion and written in standard English?

Reviewer #1: Yes

Reviewer #2: Yes

6. Review Comments to the Author

Reviewer #1: Dear Authors, Thank you for revising your manuscript according to the reviewers' comments. Their manuscript is now ready for publication.

Reviewer #2: Dear authors,

Thank you for carefully considering and applying my comments into the revised manuscript. The manuscript is much improved. As stated in my previous review, this manuscript covers an important topic and provides very pertinent and well-presented recommendations for animal ethics review committees. I will be excited to see this manuscript published and utilized to improve the processes involved in decision-making by animal ethics review committees.

While I feel the authors have adequately addressed, responded to, and incorporated my comments to the previous version, in this version, I noticed a few places where minor revisions are warranted. Please see below:

1. Results line 143 - I believe the number 586 should be updated to 662 to match Figure 1?

2. Results line 144 - Please spell out "eighty-three" as this number starts a sentence.

3. Results lines 144-150 - Throughout this sentence, I suggest using a softer word other than "wrong" -- perhaps "inapplicable" or "irrelevant" in place of the word "wrong." Authors - please feel free to use your discretion here, this is just a suggestion!!

4. Results line 192 - Should reference 35 be included in the first set of parentheses with references 30 and 38?

5. Results line 196 - I think the semi-colon is misplaced, instead there should be a comma after the reference list (before the word "while").

6. Results line 203 - The word "in" is not needed.

7. Results line 244 - Please check references here. Should there be three or four total supporting citations/studies? Also, reference 35 is not included in this list but is referenced in the next sentence, and reference 31 is not cited again at all in this paragraph.

8. Results line 259 - Missing space between sentences.

9. Discussion line 296 - Missing period between sentences.

10. Throughout - there are few sentences with double periods at the end (e.g., lines 177, 228, 241).

7. PLOS authors have the option to publish the peer review history of their article (what does this mean? ). If published, this will include your full peer review and any attached files.

**Do you want your identity to be public for this peer review?** For information about this choice, including consent withdrawal, please see our Privacy Policy .

Reviewer #1: No

Reviewer #2: No

---

## [Author Response · Author response to Decision Letter 1]

17 Dec 2024

Dear Reviewers,

Thank you for your careful evaluation of our paper. We have now addressed all the minor issues mentioned and the changes are visible in the manuscript with tracked changes. This is also outlined in the rebuttal letter. We sincerely appreciate the work you have undertaken to improve our manuscript.

---

## [Decision Letter · Decision Letter 2]

19 Jan 2025

How Animal Ethics Committees make decisions – a scoping review of empirical studies

PONE-D-24-34445R2

Dear Dr. Milford,

We’re pleased to inform you that your manuscript has been judged scientifically suitable for publication and will be formally accepted for publication once it meets all outstanding technical requirements.

Kind regards,

Jayonta Bhattacharjee

Academic Editor

PLOS ONE

Additional Editor Comments (optional):

Reviewers' comments:

Reviewer's Responses to Questions

**Comments to the Author**

1. If the authors have adequately addressed your comments raised in a previous round of review and you feel that this manuscript is now acceptable for publication, you may indicate that here to bypass the “Comments to the Author” section, enter your conflict of interest statement in the “Confidential to Editor” section, and submit your "Accept" recommendation.

Reviewer #2: All comments have been addressed

2. Is the manuscript technically sound, and do the data support the conclusions?

Reviewer #2: Yes

3. Has the statistical analysis been performed appropriately and rigorously? 

Reviewer #2: N/A

4. Have the authors made all data underlying the findings in their manuscript fully available?

Reviewer #2: Yes

5. Is the manuscript presented in an intelligible fashion and written in standard English?

Reviewer #2: Yes

6. Review Comments to the Author

Reviewer #2: Dear authors,

Thank you for making the requested revisions. This manuscript is now ready for publication.

7. PLOS authors have the option to publish the peer review history of their article (what does this mean? ). If published, this will include your full peer review and any attached files.

**Do you want your identity to be public for this peer review?** For information about this choice, including consent withdrawal, please see our Privacy Policy .

Reviewer #2: No

---

## [Editor Report · Acceptance letter]

PONE-D-24-34445R2

PLOS ONE

Dear Dr. Milford,

I'm pleased to inform you that your manuscript has been deemed suitable for publication in PLOS ONE. Congratulations! Your manuscript is now being handed over to our production team.

Kind regards,

on behalf of

Dr. Jayonta Bhattacharjee

Academic Editor

PLOS ONE